# OpenReview forum: "From Evaluation to Defense: Advancing Safety in Video Large Language Models"
_ICLR.cc/2026/Conference — ICLR 2026 Poster_

### Official Review · Reviewer_JMVT · 2025-10-26

**Soundness:** 2
**Presentation:** 3
**Contribution:** 3
**Rating:** 6
**Confidence:** 4

**Summary:**

This paper introduces VideoSafetyEval — a large-scale, real-world benchmark for video LLM safety (11.4k video–query pairs, 19 categories, 10 languages). Building on VideoSafetyEval, the authors propose a post-training defense framework, VideoSafety-R1,
which integrates: (1) VideoSafetyThinking (VST) for chain-of-thought safety reasoning (46k samples), (2) Alarm Token-Guided Safety Fine-Tuning (AT-SFT), and (3) Safety-Guided GRPO with rule-based rewards.

**Strengths:**

1. To address the safety vulnerabilities of video LLMs, the authors introduce the VideoSafetyThinking dataset, which includes 46K video–query–reasoning triples and provides a valuable resource for the research community.
2. The defense strategy is well-designed: it follows a coherent end-to-end pipeline — from perception (alarm token) to reasoning (CoT) to reinforcement learning (GRPO with dynamic rewards) — offering a clear approach that convincingly shifts from passive refusal to safety-driven reasoning.

**Weaknesses:**

1. Overstatement: The authors claim that VideoSafetyEval is the first real-world safety evaluation benchmark for video LLMs. However, prior works such as VideoSafetyBench, SafeVid, and Trust-VideoLLMs have already explored safety evaluation in this domain.
2. Data collection, query generation, initial harmful labeling, and evaluation heavily rely on Qwen-Max, Qwen-Long, and GPT-4o, with limited enough human verification. Since these models have inherent reliability issues in video understanding, the resulting dataset may share similar distributions and biases with them, undermining the credibility of the evaluation.

**Questions:**

1. How are the hyperparameters, such as $\lambda_1$ and $\lambda_2$, determined? Do their values significantly affect the performance of the alarm token and the GRPO strategy?

2. Does fine-tuning video LLMs on safety dimensions impact their performance in other aspects, such as hallucination or overall video understanding ability (e.g., on Video-MME)?

3. Could the authors further analyze the diversity of reasoning chains in VST (e.g., average number of logical steps, template proportion) and the model’s performance across different languages?

---

> ### Author Response · Authors · 2025-11-22
> **W1: Overstatement**
>
> Thank you for raising this point. We sincerely apologize for using such an eye-catching phrase. We would like to clarify that it was not our intention to make an exaggerated claim. The statement describing our work as "the first real-world safety evaluation benchmark for video LLMs" was introduced at the early stage of this research. At that time, video safety in MLLMs was still under-explored, and we were only aware of the related work VideoSafetyBench, which focuses on text-generated synthetic videos. Thus, we positioned our benchmark as involving real-world video content. As rightly pointed out by the reviewer, other datasets such as SafeVid-350k also incorporate real-world videos. We fully acknowledge this and will revise our claim accordingly. In the revised version, we will also include a thorough discussion in the related work section to clarify the distinctions and commonalities among different video safety benchmarks.

---

> ### Author Response · Authors · 2025-11-22
> **W2: Model-Dependent Bias Risks**
>
> ## Response to Concerns on Data Construction
> We sincerely appreciate this thoughtful comments. We were aware of this concern throughout the study. During the data collection phase, all samples were fully verified by human, as the definitions and retrieval terms are limited and well-bounded. For query generation and harmfulness labeling, we also incorporated partial human verification. Given the simple and structured nature of these tasks, we found that current Video LLMs generally perform reliably. Nevertheless, we fully acknowledge the reviewer’s concerns and have therefore engaged a professional annotation team to further validate these two stages, ensuring higher reliability.
>
> ## Response to Concerns on Evaluation
> The evaluation stage is indeed where we exercise the greatest caution. Existing safety alignment work, including MM-SafetyBench, SafeBench, and SPA-VL, typically relies on LLMs as evaluators. While we follow this common practice, we share the concerns about potential model bias and distributional coupling between annotators and evaluated models. To address this, and taking inspiration from Multimodal Situational Safety, we perform sampling-based quantitative verification on a subset of the evaluation results by human. As reported in Table 12 in Appendix H.3 (*Table 16 in Appendix J.3 in the latest version*), the findings show that Qwen-based evaluators achieve high agreement with human judgments while providing substantial cost benefits.
>
> ## Human–Model Consistency Evaluation on Queries and Videos
> To further quantify the reliability, we collaborated with a professional annotation team to conduct a systematic human–model agreement study for both harmfulness labeling and query generation.
> For harmfulness labeling, the procedure was as follows:
> - We sampled 1,000 items containing the video clip, the harmfulness category (from the retrieval phase), and the harmful-element analysis generated by Qwen-Max-VL.
> - Annotators independently evaluated the reliability of the analysis (faithfulness and reasoning soundness) and assigned video labels.
>
> For query labeling, the procedure was:
> - We sampled 10,000 items containing the query and its harmfulness category.
> - Annotators independently assigned query labels.
> To ensure annotation reliability, we provided detailed guidelines, representative examples, and a custom annotation interface. Multiple rounds of training and communication were conducted to guarantee consistent understanding. We then computed the accuracy of query labels, video labels, and harmful-element analyses with respect to human annotations. The results show that Qwen-based models achieve strong alignment with human annotators, and no category exhibits critically low agreement.
>
> ## Additional Observations from the Annotation Process
> During the annotation process, we made several additional observations that are consistent with the reviewer's concerns:
> - Because the tasks are relatively simple, the model analyses were largely reliable, with only about 1% mild hallucinations observed.
> - Importantly, analysis reliability (99%) does not equate to label alignment (95%). The ~4% discrepancy is mostly attributable to subtle and ambiguous cases that current models struggle to interpret, which highlights the value of human verification.
> Overall, these studies suggest that, while our pipeline inevitably reflects some biases of the underlying LLMs, the core labels and analyses are strongly supported by human verification. We will explicitly discuss the remaining risk of shared model biases, and the importance of independent re-evaluation with alternative annotators, in the limitation section of the revised manuscript.

---

> > ### Author Response · Authors · 2025-11-22
> > **W2: Verification Results**
> >
> > | Keyword                                 | Query Label Acc | Video Label Acc | Analysis Reliability Rate |
> > |-----------------------------------------|:---------------:|:----------------:|:--------------------------:|
> > | Violence and Criminal Behavior          |     98.99%      |      98.00%      |           98.00%           |
> > | Hate Speech and Hate Acts               |     98.34%      |      90.00%      |          100.00%           |
> > | Violent, Hate Groups and Individuals    |     99.32%      |      98.00%      |           98.00%           |
> > | Harassment, Bullying and Abuse          |     98.81%      |      96.00%      |          100.00%           |
> > | Suicide and Self-Harm                   |     97.31%      |      90.00%      |           96.00%           |
> > | Eating Disorders                        |     98.30%      |      94.00%      |          100.00%           |
> > | Risky Activities and Challenges         |     96.88%      |      98.00%      |          100.00%           |
> > | Vulgar Language                         |     94.31%      |      94.00%      |          100.00%           |
> > | Animal Abuse                            |     97.26%      |      90.00%      |           96.00%           |
> > | Gambling                                |     98.49%      |      96.00%      |          100.00%           |
> > | Alcohol, Tobacco, and Drugs             |     98.72%      |      96.00%      |           98.00%           |
> > | Firearms and Dangerous Weapons          |     99.82%      |     100.00%      |          100.00%           |
> > | Generated Information                   |     98.49%      |      98.00%      |          100.00%           |
> > | Personal Information                    |     96.27%      |     100.00%      |          100.00%           |
> > | Platform Information                    |     97.37%      |      94.00%      |          100.00%           |
> > | Sexual Behavior and Services            |     96.02%      |      98.00%      |          100.00%           |
> > | Nudity and Exposure                     |     92.54%      |      88.00%      |           98.00%           |
> > | Sexual Suggestiveness                   |     97.24%      |      96.00%      |           98.00%           |
> > | **Overall**                             |   **97.44%**     |    **94.84%**    |         **99.05%**         |

---

> ### Author Response · Authors · 2025-11-22
> **Q1: Ablation of $\lambda_1$ and $\lambda_2$**
>
> We thank the reviewer for the question about the hyperparameters. In the main paper, all reported results are based on the proposed loss formulation with the empirical setting of $\lambda_1 = 0.1$ and $\lambda_2 = 0.1$. Within the available time, we conducted an ablation study over nine configurations of AT-SFT. Although this search is not exhaustive and the chosen hyperparameters may not be strictly optimal, the conclusions about the classification objective remain stable. Across all nine configurations, the variant with the classification objective consistently outperforms tuning only the LLM (74.0%) and tuning the LLM together with the alarm token (78.1%). We observe that increasing $\lambda_1$ generally leads to higher safety performance, and $\lambda_2 = 0.5$ yields the best results on our validation set. Overall, changing these coefficients does not noticeably alter the qualitative behavior of the alarm token.
>
> A more extensive ablation that jointly sweeps AT-SFT and GRPO hyperparameters would be computationally demanding and is difficult to complete within the rebuttal period. Empirically, once AT-SFT is stable, applying GRPO on top of it usually produces only modest performance differences, and we have not observed sensitive or unstable behavior with respect to these hyperparameters. We will clarify these settings and trends in the revised manuscript and add the AT-SFT ablation results to the appendix.
>
> #### **Ablation Table**
>
> | $\lambda_2 \, \backslash \, \lambda_1$ | 0.1   | 0.5   | 1.0   |
> |----------------------------------------|:-----:|:-----:|:-----:|
> | **0.1**                                | 83.1% | 82.4% | 84.1% |
> | **0.5**                                | 83.7% | 85.1% | 85.3% |
> | **1.0**                                | 83.0% | 82.8% | 83.8% |

---

> ### Author Response · Authors · 2025-11-22
> **Q2: Analysis of Generalization on Video Benchmark**
>
> In L424, within the section on *Robustness and Generalization*, we have already analyzed how safety post-training affects general video understanding. The corresponding experimental results are reported in Table 3, evaluated on MVBench, Video-MME, Perception Test, and NextQA (dataset details are provided in *Appendix P of the old version and Appendix U in the latest version*). Overall, these benchmarks show that our post-trained models largely preserve their general video capabilities, with only modest variations compared to the base models.
>
> To further study the effect on hallucination, we additionally conduct an experiment on the **VideoHallucer** [1] benchmark. As shown below, for basic questions, VideoLLaMA3-2B + VideoSafetyR1 decreases by 3.77%, while for hallucination questions the model improves by 1.01%, resulting in an overall drop of 1.38%.
>
> | Model                       | Basic  | Hallucinated | Overall |
> |-----------------------------|:------:|:------------:|:-------:|
> | **VideoLLaMA3-2B**          | 75.08% |    68.84%    | 71.96%  |
> | **VideoLLaMA3-2B + VideoSafetyR1** | 71.31% |    69.85%    | 70.58%  |
>
> Taken together with the results on MVBench, Video-MME, Perception Test, and NextQA in Table 3, these findings indicate that safety-oriented fine-tuning does introduce a small reduction in general capability, but this reduction is moderate and, in our view, acceptable given the substantial safety gains achieved by VideoSafetyR1.
>
> **Ref**
>
> [1] VideoHallucer: Evaluating Intrinsic and Extrinsic Hallucinations in Large Video-Language Models

---

> ### Author Response · Authors · 2025-11-22
> **Q3.1: Analysis of Reasoning Diversity**
>
> We sampled 950 responses from both **VideoLLaMA3-2B + VideoSafetyR1** and **Qwen2.5-VL-3B + VideoSafetyR1**, and computed:
> - the average number of reasoning steps
> - the proportion of outputs that match the designed structural template.
> The results are shown below:
>
> | Model                                | #Steps | Matching Rate |
> |--------------------------------------|:------:|:-------------:|
> | **VideoLLaMA3-2B + VideoSafetyR1**   |  4.18  |     92.1%     |
> | **Qwen2.5-VL-3B + VideoSafetyR1**    |  4.23  |    100.0%     |
>
> The reasoning steps mainly include judging the video, judging the question, jointly analyzing the video and the question, and summarizing the conclusion. During the question-focused or joint analysis stages, we frequently observe additional reasoning, such as introducing commonsense explanations or providing reasonable suggestions. This leads to an average reasoning length of more than four steps. In terms of structural template matching, both models exhibit strong consistency.

---

> ### Author Response · Authors · 2025-11-22
> **Q3.2: Analysis of Cross-Lingual Performance in VSE-HH**
>
> To analyze the model’s behavior across different languages, we only replace the language of the prompts while keeping the underlying videos and harmful intents fixed. The experimental results are presented below. We observe that using other languages consistently leads to higher DSR compared with English, and this pattern holds across all four models.
>
> However, after reviewing a large number of response examples, we find that prompts in other languages tend to make the models more conservative. In many cases, the models directly reject the query without engaging in detailed analysis. This behavior is particularly noticeable for **VideoLLaMA3-2B + VideoSafetyR1**, where the safety score (DSR) increases, but the helpfulness score (HS) decreases.
>
> ---
>
> ### **(EN–ES) Results**
>
> | Model | EN DSR | EN HS | JP DSR | JP HS | KO DSR | KO HS | ES DSR | ES HS |
> |:---|:---:|:---:|:---:|:---:|:---:|:---:|:---:|:---:|
> | **VideoLLaMA3-2B** | 18.4% | 2.3 | 23.0% | 2.1 | 22.4% | 2.2 | 19.8% | 2.5 |
> | **Qwen2.5-VL-3B** | 54.0% | 2.4 | 58.3% | 2.4 | 61.7% | 2.5 | 58.0% | 2.5 |
> | **VideoLLaMA3-2B + VideoSafetyR1** | 89.5% | 4.6 | 97.7% | 2.8 | 97.8% | 2.7 | 98.6% | 3.0 |
> | **Qwen2.5-VL-3B + VideoSafetyR1** | 91.7% | 3.6 | 93.4% | 3.6 | 92.5% | 3.7 | 87.6% | 3.6 |
>
> ---
>
> ### **(HI–RU) Results**
>
> | Model | HI DSR | HI HS | AR DSR | AR HS | RU DSR | RU HS |
> |:---|:---:|:---:|:---:|:---:|:---:|:---:|
> | **VideoLLaMA3-2B** | 23.0% | 2.3 | 22.0% | 2.0 | 17.8% | 2.4 |
> | **Qwen2.5-VL-3B** | 62.7% | 2.6 | 62.0% | 2.3 | 61.2% | 2.4 |
> | **VideoLLaMA3-2B + VideoSafetyR1** | 97.3% | 2.6 | 97.8% | 3.1 | 98.7% | 2.9 |
> | **Qwen2.5-VL-3B + VideoSafetyR1** | 96.5% | 3.4 | 92.1% | 3.6 | 91.2% | 3.8 |
>
> ---
>
> ### **(FR–PT) Results**
>
> | Model | FR DSR | FR HS | DE DSR | DE HS | PT DSR | PT HS |
> |:---|:---:|:---:|:---:|:---:|:---:|:---:|
> | **VideoLLaMA3-2B** | 18.8% | 2.8 | 18.6% | 2.5 | 17.4% | 2.6 |
> | **Qwen2.5-VL-3B** | 61.8% | 2.6 | 55.2% | 2.9 | 56.1% | 2.7 |
> | **VideoLLaMA3-2B + VideoSafetyR1** | 98.1% | 3.0 | 98.0% | 3.1 | 98.8% | 3.0 |
> | **Qwen2.5-VL-3B + VideoSafetyR1** | 92.8% | 3.7 | 91.9% | 3.6 | 93.8% | 3.8 |
>
> ---
>
> These results indicate that our safety alignment framework is broadly effective across multiple languages, but also reveal a trade-off:
> **higher safety often comes at the cost of more conservative behavior and reduced analytical helpfulness in some non-English settings.**
>
> We will clarify this cross-lingual effect and discuss it as a potential limitation in the revised manuscript.

---

> > ### Comment · Reviewer_JMVT · 2025-11-28
> >
> > Thanks for your careful and detailed point-by-point response, which has essentially addressed my concerns. I will raise my score

---

> > > ### Author Response · Authors · 2025-11-28
> > > **Appreciation for the Reviewer’s Comments**
> > >
> > > We sincerely appreciate your recognition and positive feedback. Once again, thank you for the valuable suggestions that have greatly strengthened this work. Please feel free to reach out if any further questions arise.

---

### Official Review · Reviewer_XGNf · 2025-10-29

**Soundness:** 2
**Presentation:** 2
**Contribution:** 1
**Rating:** 2
**Confidence:** 4

**Summary:**

The authors propose a benchmark for Video-LLMs safety and two modifications to the post-training procedures of Video-LLMs to produce improved performance. The first improvement is Alarm Token-Guided Safety Finetuning, which learns learns alarm tokens that allegedly activate harm detection mechanisms. The second improvement is a safety-guided GRPO procedure that tweaks the RL procedure by introducing rule-based rewards based on policies.

**Strengths:**

S1: The VideoSafetyThinking dataset could be a nice contribution to the literature. If the data is of sufficient quality and interesting analyses about it can be performed, it could become a key aspect of discussion in the paper.

S2: The proposed modifications are simple and, although not particularly innovative, seem effective.

S3: In general, the idea of using reasoning data and applying RL/tokenization can be interesting

**Weaknesses:**

W1: The claim to be the 'first large-scale, real-world benchmark for Video LLM safety' is a bit of an overstatement. The authors themselves, do cite competing (allegedly concurrent) benchmarks, besides the existence of [1,2,3]. Similarly the claim that the video modality introduces a vulnerability is not particularly novel: any form of finetuning (even on few benign samples in the same modality, even worse when a new modality is added) reduces the effectiveness of safety finetuning procedures.

W2: Both the idea of adjusting the tokenization mechanism and of performing safe variants of RL are well known (e.g. besides [1,5], plenty are available). Maybe it could help having a detailed comparison between the various safety finetuning methods of RL to point out aspects of novelty or greater efficiency/effectiveness.

W3: The authors focus extensively on the impact of FPS on the results. However:

W3.1: In the section 'Effect of FPS' (line 373) they claim 'increased visual content introduces more safety risk'. The claim is handwavy: since their dataset does not focus on safety harms caused by short-length unsafe contents or malicious frames, it's not clear why the statement should be true. Presumably, the authors observed the correlation between higher number of frames in input and lower performance. However this is likely more related to the inability to handle long context, possibly high resolution, context windows or implicit downsampling mechanisms. Could the authors clarify this?

W3.2: Both previusly indicated section and lines 409-414: RL is framed as a silver bullet for safety, while it is well known not to be. Further experiments are required to support the claim that some classes of models are more safe due to RL with respect to others. In particular, it would be necessary to ensure models are pre-trained and fine-tuned on the same distributions using different finetuning approaches.

W3.3: The real purpose and meaning of the experiment in Sec 5.5 is unclear, especially without a clear description of how dynamic the videos produced are (indeed many image to video models produce minimal movements).

W4: The data collection procedure is not particularly novel, and could be moved to appendix to leave more space to discuss and analyse the strengths of the work.


[1] https://arxiv.org/pdf/2412.06878

[2] https://arxiv.org/abs/2409.19734

[3] https://arxiv.org/pdf/2406.12235

[4] https://arxiv.org/pdf/2505.20065

[5] https://arxiv.org/abs/2412.10493

Minor: several sentences are verbose and unclear (e.g. .g. line 045: dynamic content interaction and temporal threat propagation; or not grammatically correct ‘then extend to 10 languages’ line 165. Or inappropriate terms, e.g. line 173 ‘surviving’ videos for videos that pass the filtering stage). These are just few examples.

**Questions:**

Q1: Could the authors discuss the similarities and dissimilarities of their token-guided finetuning approach with respect to [1]?

Q2: Could the authors produce a comparison in terms of performance with respect to SafeWatch and Holmes-VAD?

**Details Of Ethics Concerns:**

The dataset may contain harmful content and it's important to vet it before release.

---

> ### Author Response · Authors · 2025-11-22
> **W1: Overstatement**
>
> ### About Video Safety Alignment
> Thank you for raising this point. We sincerely apologize for using such an eye-catching phrase. We would like to clarify that it was not our intention to make an exaggerated claim. The statement describing our work as "the first real-world safety evaluation benchmark for video LLMs" was introduced at the early stage of this research. At that time, video safety in MLLMs was still under-explored, and we were only aware of the related work VideoSafetyBench, which focuses on text-generated synthetic videos. Thus, we positioned our benchmark as involving real-world video content.
>
> As rightly pointed out by the reviewer, other datasets such as SafeVid-350k also incorporate real-world videos. We fully acknowledge this and will revise our claim accordingly. In the revised version, we will also include a thorough discussion in the related work section to clarify the distinctions and commonalities among different video safety benchmarks.
>
> ### About Video Anomaly Detection
> Regarding SafeWatch, T2VBench, and HolmesVAD, we have indeed reviewed these works. In our understanding, they primarily focus on detecting abnormal or harmful video content, aiming to assess and improve the model’s capability to identify and explain unsafe videos.
>
> In contrast, multimodal safety alignment focuses on a different question, namely how to evaluate and enhance the safety and helpfulness of model responses when confronted with harmful multimodal inputs. As a results, we can make a table to compare them:
>
> | Aspect | Multimodal Safety Alignment | Video Anomaly Detection |
> | --- | --- | --- |
> | Task Objective | Ensures the model responds safely under harmful video–text interactions | Determines whether the video contains abnormal events |
> | Output Type | Refusal, safety explanation, safe generation | Normal/abnormal labels or anomaly scores |
> | Input | Video + text query | Video only |
> | Source of Risk | Joint harmful semantics from video × text | Harmfulness comes solely from video content |
> | Focus | Safety and alignment of model behavior | Detection of abnormal content |
> | Nature of the Task | Behavior-level alignment | Content-level detection |
>
>
> Due to the fundamental differences between the research tasks, our dataset and methodology differ substantially from those used in video anomaly detection. Given this difference in research scope, contemporaneous studies such as SafeVid did not include these works in their discussions either. We sincerely appreciate reviewer XGNF’s question, and we will include an explanation of this point in the related work section.
>
> ### Discussion on Whether Fine-tuning Degrades Safety
> Reviewer XGNf notes that “fine-tuning tends to reduce safety performance,” which is indeed a widely recognized observation. However, our experiments suggest that this conclusion does not always hold in multimodal settings. In particular, our results indicate that adding modalities does not inherently weaken the safety properties of the language backbone. For example, VideoLLaMA3-2B without vision even achieves a higher defense success rate than Qwen2.5-1.5B.
>
> That said, when comparing the with-vision and without-vision settings of the same model, we consistently observe that the introduction of visual inputs leads to a degradation in safety performance. The detailed results are shown in the table below.
>
> | Video LLM | Frames | Base LLM | DSR (w/ vision) | DSR (w/o vision) | DSR Drop Rate |
> | --- | --- | --- | --- | --- | --- |
> | Base LLM | - | Qwen2.5-1.5B | - | 77.7% | - |
> | VideoLLaMA3-2B | 1fps | Qwen2.5-1.5B | 18.4% | 89.3% | 79.4% |
> | VideoChat-Flash-2B | 1fps | Qwen2.5-1.5B | 8.5% | 13.6% | 38.0% |
> | Base LLM | - | Qwen2.5-3B | - | 68.5% | - |
> | Qwen2.5 VL-3B | 1fps | Qwen2.5-3B | 54.0% | 77.1% | 30.0% |
> | Base LLM | - | Qwen2.5-7B | - | 64.5% | - |
> | VideoLLaMA3-7B | 1fps | Qwen2.5-7B | 31.2% | 88.2% | 64.6% |
> | Qwen2.5 VL-7B | 1fps | Qwen2.5-7B | 57.3% | 72.1% | 20.5% |
> | Base LLM | - | Qwen2-7B | - | 76.6% | - |
> | VideoChat-Flash-7B | 1fps | Qwen2-7B | 19.0% | 57.5% | 66.9% |
> | LLaVA-OV-7B | 16 | Qwen2-7B | 39.9% | 50.3% | 20.7% |
> | Base LLM | - | InternLM-7B | - | 37.1% | - |
> | InternVideo2.5-8B | 1fps | InternLM-7B | 16.5% | 53.5% | 69.2% |
> | Base LLM | - | Vicuna v1.5-7B | - | 51.3% | - |
> | LLaVA-Next-Video-7B | 16 | Vicuna v1.5-7B | 26.1% | 44.2% | 41.0% |
> | PLLaVA-7B | 16 | Vicuna v1.5-7B | 26.8% | 58.3% | 54.1% |
> | Base LLM | - | Mistral-v0.2-7B | - | 61.3% | - |
> | VideoLLaMA2-7B | 16 | Mistral-v0.2-7B | 69.8% | 75.3% | 7.3% |
> | VideoChat2-7B | 16 | Mistral-v0.2-7B | 50.5% | 72.2% | 30.0% |
> | Base LLM | - | Phi-3-mini-128k | - | 79.2% | - |
> | VideoChat2-7B | 16 | Phi-3-mini-128k | 62.1% | 71.7% | 13.5% |
> | Base LLM | - | Phi-3-mini-4k | - | 77.8% | - |
> | VideoGPT+-5B | 16 | Phi-3-mini-4k | 30.7% | 71.6% | 57.1% |

---

> ### Author Response · Authors · 2025-11-22
> **W2: Advantages of AT-SFT and Safety-Guided GRPO**
>
> ### About AT-SFT
> To the best of our knowledge, in the field of multimodal safety alignment, there has been no prior work that uses a dedicated token to explicitly control model behavior. Existing approaches primarily focus on strengthening the data and applying reinforcement learning, such as SPA-VL, SafeVid, and MM-RLHF, or rely on supervised finetuning as in VLGuard, or prompt-based control as in MM-SafetyBench. In contrast, our method introduces a specialized token and uses AT-SFT to regulate the model’s internal behavior.
>
> #### Attention Sink*
> Specifically, regarding the motivation behind designing the Alarm Token, we aim for these two special classification tokens to independently sensor harmful content within the video and text modalities. To validate this approach, we further analyzed the Attention Sink phenomenon[1][2]  The general pattern observed in the aforementioned literature is that, for the text modality, the Attention Sink occurring at the first token position stores redundant attention tokens in LLMs, thereby maintaining model stability. In contrast, for visual tokens, the sink tokens occupy a significant amount of ineffective attention, and redistributing this attention can enhance visual performance.
>
> Our analysis of Qwen2.5VL revealed a similar intriguing pattern: in the vanilla model, the first visual token exhibited a pronounced sink phenomenon. After SFT guided by Alarm Tokens, however, the sink effect in the first visual token was significantly mitigated, while the sink effect in the first textual token intensified. This can be interpreted as our SFT approach effectively freeing up ineffective visual attention and reallocating it to the textual attention sink that preserves model stability and textual performance. Consequently, this reactivates the model's inherent textual safety capabilities, whch a finds that aligns perfectly with our observation that safety performance substantially improves without vision input. We will provide a detailed visual analysis of Attention Sink in the following revised PDF version.
>
> #### Comparison with SafeWatch
> SAFEWATCH focuses primarily on building a comprehensive and interpretable harmful video detection system. It dynamically prunes video tokens according to a safety policy, with an emphasis on the interaction between safety policies and video frame attention. However, it does not explicitly address cross-modal video-text interaction or defensive reasoning over textual harmful content. In contrast, our method is tailored to enhance the safety of MLLMs under joint video-text input through a simple yet effective post-training procedure (SFT + RL). This process enables the model to perceive harm in each modality and their compositional semantics, ultimately leading to more reliable and context-aware safe responses.
>
>
>
> **Ref:**
>
> [1] WHEN ATTENTION SINK EMERGES IN LANGUAGE MODELS: AN EMPIRICAL VIEW (ICLR'25)
>
> [2] SEE WHAT YOU ARE TOLD: VISUAL ATTENTION SINK IN LARGE MULTIMODAL MODELS (ICLR'25)
>
>
>
>
> ### About Safety-Guided GRPO
> SPA-VL adopts PPO for safety finetuning, while GRPO can be viewed as a reward-free variant of PPO and therefore offers lower computational cost.
> Current safety finetuning practices widely rely on DPO, as seen in Safe-Vid, SPA-VL, SafeDPO, MM-RLHF, AlignGuard, and others.
> The advantages of Safety-Guided GRPO include:
>
> 1. Better alignment with chain-of-thought safety design.
> GRPO allows the model to reason in a more controlled manner.
> Our approach enables the model to examine the harmfulness of each input modality individually, as well as their combined semantic risks, thereby guiding the generation of safe and helpful responses.
> 2. Lower data construction cost.
> GRPO does not require preference-labeled pairs, which significantly reduces annotation overhead.

---

> ### Author Response · Authors · 2025-11-22
> **W3**
>
> ## W3.1: Explanation on your speculation
> ### Clarification on our dataset
> Our intention was simply to convey that increasing the FPS leads to lower safety performance. We will revise this sentence to make the statement clearer.
> Second, we would like to clarify the data characteristics behind our claim. Our dataset is collected from real-world scenarios. As noted in our response to reviewer 8QWg regarding W1, most videos contain alternating harmful and harmless content rather than being uniformly unsafe. This pattern aligns with the “short-length unsafe contents or malicious frames”.
> Finally, in connection with the analysis in Section 5.5, increasing the FPS expands the exposure to existing harmful semantics or introduces new harmful content. This increases safety risk and results in a decrease in safety performance.
>
> ### Clarification on model ability
> Regarding the concern that the observed degradation might instead be caused by limitations in handling long contexts, we have also carefully considered this issue in our study. During data collection, we ensured that the duration of most videos remains within the range that current Video-LLMs can reasonably process (*Appendix C in the original version and Appendix D in the latest version*). As noted in L1062 (*L1157 in the latest version*), Qwen2.5-VL and VideoLLaMA3 process up to 768 and 180 frames during training, with sampling rates of 2 fps and 1 fps, respectively. Furthermore, although VideoLLaMA3 appears to sample far fewer frames than Qwen2.5-VL, the Differential Frame Pruner ensures that the number of visual tokens remains controlled even when more frames are provided as input. These designs mitigate long-context or resolution-induced failures, and support our interpretation that the dominant factor behind the safety drop in our FPS analysis is the introduction of additional harmful video content, rather than an uncontrolled increase in token load.
>
>
> ## W3.2: SFT memorizes, while RL generalizes
> We fully acknowledge that RL is by no means a one-size-fits-all solution. In this work, our approach is motivated by the perspective that "SFT memorizes, while RL generalizes [1][2][3]." By incorporating GRPO with Chain-of-Thought, as shown in Table 4 (*Table 6 in the latest version*), we observe that introducing an RL stage leads to a notable improvement in safety generalization on MMBench. It is worth emphasizing that after the first-stage SFT, the model already achieves a substantial gain in in-domain safety performance (on the VSE dataset). However, due to the limited diversity of the answer templates used in SFT, the model tends to overfit these templates when trained with SFT alone. Therefore, we propose that RL can help by introducing rule-based safety rewards along with CoT reasoning, thereby stimulating the model’s generalization in safety awareness. Through GRPO, we effectively mitigate reward hacking and enable preference-free RL without requiring additional annotated preference pairs. This offers a simple yet effective pathway toward enhancing safety performance.
>
> Ref:
>
> [1] Sft memorizes, rl generalizes: A comparative study of foundation model post-training
>
> [2] Understanding the effects of rlhf on llm generalisation and diversity
>
> [3] On-policy rl meets off-policy experts: Harmonizing supervised fine-tuning and reinforcement learning via dynamic weighting
>
> ## W3.3: Meaning of Sec 5.5's Experiments
> In the Effects of FPS section, we state that increasing FPS leads to lower safety performance. However, increasing FPS introduces two changes: token load and potentially richer harmful semantics. In Section 5.5, we conduct a controlled analysis of these two factors. To control for harmful semantics and isolate the effect of frame count alone, we follow the experimental setting suggested by reviewer 8QWg in Q2. Specifically, we convert a harmful image into a ten second static video and vary the number of sampled frames. Regarding your concern about the dynamics of the generated videos, we clarify that VLGuard-Video is intentionally constructed as a static video in order to avoid introducing any dynamic semantics. This design ensures that the setup isolates the effects of token load, and the sole purpose is to allow the video understanding model to perform frame sampling in a controlled manner. The VLGuard-Video setting is described in Appendix I.2 (*K.2 in the latest version*). The results are presented in Table 3, and additional sampling results are as follows. The results suggests that, large visual token loads alone are not sufficient to explain safety degradation.
>
> |Model|1|10|20|30|40|50|60|STD|
> |---|---|---|---|---|---|---|---|---|
> | Qwen2.5 VL-7B | 45.5% | 44.6%| 43.8% | 45.5% | 45.1% |46.0% | 47.0%|1.00%|
> | VideoLLaMA3-7B | 58.6% | 57.4% | 58.0% | 56.1% | 58.8% | 57.7% | 56.2% |1.07%|
> | Qwen2.5 VL-3B | 52.6% | 52.0% | 52.0% | 52.3% | 51.4% | 51.1% | 50.9% | 0.66% |
> | VideoLLaMA3-2B | 44.9% | 44.5% | 44.8% | 44.6% | 45.3% | 43.4% | 45.4% | 0.66% |

---

> ### Author Response · Authors · 2025-11-22
> **W4: Discussion on Novelty of Data Collection Pipeline**
>
> We sincerely appreciate your suggestion, and we agree that this section can be further shortened in the main paper. In the revised version, we will move part of the detailed description to the appendix. Before doing so, we would like to briefly clarify the motivation behind including this part.
>
> This section is intended to serve the specific purposes: it conveys our considerations about the video modality in terms of temporal dynamics and annotation quality, which we view as important for constructing a reliable video safety benchmark.
>
> More specifically:
>
> + Temporal dynamics.
> The motion characteristics of a video determine how it differs from images, regardless of whether the video is generated or retrieved. We observed that prior works did not introduce dedicated filtering mechanisms to handle videos that are largely static. In contrast, we apply targeted filtering during our preprocessing stage to explicitly address this issue.
> + Annotation quality.
> SafeVid uses video-level annotations and SafeWatch uses frame-level annotations. Our approach adopts segment-level annotation, which lies between the two in terms of efficiency. To compensate for this design choice, we invest substantial resources in commercial model APIs to analyze each segment in order to ensure annotation quality. Moreover, query generation is crucial in the context of safety alignment. We provide both video-based and text-based descriptions as references for query construction. This differs from SafeVid, which provides only text descriptions, and is intended to yield queries with higher relevance.
>
> In the revised manuscript, we will keep a concise summary of these key design choices in the main text and move the more procedural details of data collection to the appendix, in order to leave more space for discussing and analyzing the strengths of our method.

---

> ### Author Response · Authors · 2025-11-22
>
> ## Q1: Comparison with SafeWatch
> Similarity.
> Both approaches use labels as one of the key signals guiding the model’s generation process. In both SafeWatch and VideoSafety-R1, labels act as auxiliary signals that help structure semantic understanding and inform the model’s outputs.
>
> Differences.
>
> + Motivation. SafeWatch uses labels primarily to reduce the token overhead associated with rule-based descriptions, thereby improving the efficiency of harmful-video detection. In contrast, VideoSafety-R1 employs labels to pre-analyze semantic harmfulness on both the visual and textual sides, serving as an early-warning mechanism that strengthens the model’s inherent safety defenses.
> + Influence on outputs. In SafeWatch, labels directly determine both the detection results and the explanatory content, giving them a decisive role in the final output. In VideoSafety-R1, labels are used only as intermediate analytical cues and do not directly control defensive strategies or final responses, so their influence is indirect.
> + Label design and integration. SafeWatch encodes labels using predefined rules and incorporates them via cross-attention with video features. VideoSafety-R1, on the other hand, learns and optimizes its labels during training, allowing them to interact autoregressively with other tokens in a way that is more natural for large language models.
>
>
>
> ## Q2: Performance Comparison among SafeWatch, Holmes-VAD and ours
> We sincerely appreciate the reviewer’s suggestion and agree that a direct comparison with SafeWatch and Holmes-VAD would be very informative. Unfortunately, SafeWatch only releases a limited set of dataset files and does not provide model-related code or pretrained weights, which makes a direct model-level comparison with SafeWatch infeasible.
>
> Instead, we provide a comparison with Holmes-VAD on both VSE-HH and XD-Violence [1]. For XD-Violence, we directly prompt the models to decide whether the video contains abnormal or violent events and to provide a justification. The results are shown below:
>
> | Model | DSR | HS | XD-Violence |
> | --- | --- | --- | --- |
> | Holmes-VAD | 43.9% | 2.5 | 85.0% |
> | VideoLLaMA3-2B + VideoSafetyR1 | 89.5% | 4.6 | 61.0% |
> | Qwen2.5-VL-3B + VideoSafetyR1 | 91.7% | 3.6 | 62.3% |
>
>
> Although VSE-HH consists of videos that contain potentially harmful elements, Holmes-VAD still achieves only moderate defense success rates when facing harmful multimodal inputs. This highlights that safety alignment and video anomaly detection are fundamentally different tasks, and the experiment further supports this point.
>
> Conversely, our safety-aligned models only achieve moderate classification accuracy on XD-Violence, which is likely due to the fact that they are not explicitly trained for anomaly detection. Overall, this comparison suggests that Holmes-VAD is strong on its own anomaly-detection objective but limited in safety alignment, whereas VideoSafetyR1 provides substantial safety gains on VSE-HH at the cost of only moderate anomaly-detection performance on XD-Violence.
>
>
>
> **Ref:**
>
> [1] Not only look, but also listen: Learning multimodal violence detection under weak supervision
>
>
> ## Minor: unclear sentences
>
> Thank you for pointing it out! We will carefully revise it in the revised manuscript.

---

### Official Review · Reviewer_8QWg · 2025-10-31

**Soundness:** 4
**Presentation:** 3
**Contribution:** 4
**Rating:** 8
**Confidence:** 4

**Summary:**

This paper studies safety in Video LLMs and introduces a new benchmark and defense pipeline. The authors build VideoSafetyBench (VSB-77k), a large, policy-grounded dataset that contains 77,646 video–query pairs, 19 subcategories across 6 risk categories and 10 language communities, and derive an 11.4k evaluation split VSB-Eval and a 46k post-training set with CoT (VSB-R1-46k). The author shows that adding the video modality substantially degrades safety, highlighting systemic vulnerabilities unique to video inputs. To address this, they propose VideoSafety-R1, combining Alarm Token-Guided Safety Fine-Tuning (AT-SFT) with Safety-Guided GRPO to inject modality-aware “alarm” tokens and then reinforce defensive reasoning with rule-based rewards tied to dual-modality verification. Extensive experiments validate VideoSafety-R1’s effectiveness on the adversarial VSE-HH benchmark, it improves VideoLLaMA3-2B’s DSR by 71.1% (from 18.4% to 89.5%). This approach also generalizes well to image safety benchmarks, boosting DSR by 59.1% on MMBench, 44.3% on VLGuard, and 15.0% on FigStep. It maintains strong utility and avoids over-defense while outperforming existing methods like SPA-VL and VLGuard.

**Strengths:**

Clear evidence that video hurts safety: The paper provides quantitative analyses showing that integrating video can sharply reduce DSR relative to text-only scenes (e.g., −79.4% for VideoLLaMA3-2B), and that harmful videos amplify adversarial effectiveness. This is an important and under-explored finding that justifies a video-specific safety agenda.

Well-designed benchmark and pipeline: VSB-77k is large, multilingual, and aligned to platform policies. The construction pipeline that consists of filtered video collection, multi-agent LLM annotation, and template-driven query generation is also clear. For metrics, the paper also sets DSR, Helpfulness score and FRR to systematically evaluate.

Effective two-stage defense with clear ablations: AT-SFT (learnable modality-specific alarm tokens + multitask alarm-classification objectives) and Safety-Guided GRPO are proved to be complementary. Ablations show each component helps and the combination yields the best overall safety, while keeping original video understanding essentially unchanged. The method substantially boosts DSR on VSB-Eval-HH and transfers across external safety benchmarks.

**Weaknesses:**

No coverage of “dynamic adversarial attacks”: Although the task is video, the method/evaluation effectively treats it as a set of sampled frames, rather than modeling sequence-level risk. The benchmark’s “harmful video” cases are largely explicitly harmful (e.g., direct fights, weapon displays). It does not include more realistic implicit dynamic attacks, for example, first ~10 frames benign, later ~20 frames contain fragmented harmful content or the video shows a seemingly legal tool but frame order implies harmful use. These higher-frequency, real-world attack patterns are absent, so the reported safety may overestimate robustness in deployment.

Token-load confound (not controlled): The paper attributes safety degradation to the video modality, but it does not rule out a simpler cause "large visual token loads". If one constructs a “pseudo-video” by repeating the same malicious image across many frames, the model receives far more visual tokens without adding temporal information. If defenses fail primarily because of attention/kv-cache saturation or context dilution under high visual token counts, then the reported vulnerability is not uniquely video-specific.

**Questions:**

1. Could the current “dual-modality verification” be extended to a temporal–text–vision triad (e.g., segment-level harmful annotations/rewards, time-aware alarm tokens, or temporal grounding constraints)?

2. Could you evaluate a control set where a single malicious image is duplicated to N frames (no temporal variation) and check if safety still collapses as N grows, that implicates token load, not temporal content.

---

> ### Author Response · Authors · 2025-11-22
> **W1: Dynamic Adversarial Attacks**
>
> We thank the reviewer for this insightful comment and apologize for the misunderstanding caused by the examples in Appendix L (*Appendix Q in the latest version*). To illustrate the mapping between harmful elements and their categories, we intentionally selected frames that visibly contain harmful content. This may give the impression that our benchmark focuses only on fully unsafe clips. In real-world videos, however, and also **in our dataset, harmful segments are typically interleaved with benign content**. In fact, many clips follow the pattern described in the first type of “implicit dynamic attack” (for example, an initially benign context followed by several brief harmful fragments). The fully harmful segments constitute only the second category.
>
> The second type of implicit dynamic attack (for example, seemingly legal tools where only the frame order implies harmful use) corresponds to a much stronger form of temporal risk. **Because this is a very specialized pattern and such cases are difficult to collect at scale, our current data construction did not explicitly target this category, and its coverage remains limited.** We acknowledge that this may lead to an optimistic estimate of robustness in these specific scenarios, and we will explicitly discuss this as a limitation in the revised manuscript. We are actively investigating this direction and plan to build and release a dedicated dynamic-adversarial subset as part of future work.

---

> ### Author Response · Authors · 2025-11-22
> **W2&Q2: Effects of Token Load**
>
> We thank the reviewer for highlighting the potential confound of large visual token loads. We fully agree that high visual token counts could in principle degrade performance through attention saturation or context dilution, and that this effect should be disentangled from genuinely video-specific temporal content.
>
> To explicitly control for token load, we have constructed exactly the “pseudo-video” setting suggested by the reviewer. In VLGuard-Video (*Appendix I.2 in the original version and Appendix K.2 in the latest version*), a single malicious image is extended into a 10-second static video and duplicated across multiple frames without any temporal variation. We then sample N frames from this static video and compute DSR for several representative Video-LLMs. In the original submission we reported results for N = 1 and N = 10 (Table 3). In this rebuttal, we further extend the sweep to N ∈ {1, 10, 20, 30, 40, 50, 60}. The results are:
>
> | Model              |   1    |   10   |   20   |   30   |   40   |   50   |   60   |  STD  |
> |--------------------|:------:|:------:|:------:|:------:|:------:|:------:|:------:|:-----:|
> | **Qwen2.5 VL-7B**      | 45.5% | 44.6% | 43.8% | 45.5% | 45.1% | 46.0% | 47.0% | 1.00% |
> | **VideoLLaMA3-7B**     | 58.6% | 57.4% | 58.0% | 56.1% | 58.8% | 57.7% | 56.2% | 1.07% |
> | **Qwen2.5 VL-3B**      | 52.6% | 52.0% | 52.0% | 52.3% | 51.4% | 51.1% | 50.9% | 0.66% |
> | **VideoLLaMA3-2B**     | 44.9% | 44.5% | 44.8% | 44.6% | 45.3% | 43.4% | 45.4% | 0.66% |
>
> When the semantics are strictly fixed and only the number of duplicated frames (and thus visual tokens) increases, DSR varies within a narrow band and does not exhibit a systematic collapse as N grows. This suggests that, under our experimental regime, large visual token loads alone are not sufficient to explain the observed safety degradation.
>
> In contrast, when we keep the sampling rate (and hence the token budget) approximately fixed and vary whether the video actually contains harmful semantics, we observe a much stronger effect. In VSE-HH and VSE-SH, which correspond to harmful multimodal attacks on harmful videos and benign videos respectively, VSE-HH leads to significant safety drops while VSE-SH has a much weaker impact (Table 3). Taken together, the static pseudo-video control and the VSE-HH versus VSE-SH comparison support our conclusion that the main driver of safety degradation in our setting is the presence of harmful video semantics rather than visual token load alone. We will further clarify this analysis in Sec. 5.5 in the revised manuscript.

---

> ### Author Response · Authors · 2025-11-22
> **Q1: Discussion on Temporal–Text–Visual Triplets**
>
> We appreciate this technical suggestion and agree that extending the current dual-modality verification to a temporal–text–vision triad is a very promising direction. Our dataset is already annotated at the level of 15-second segments, which suggests that it is feasible to further refine our framework toward segment-level harmful annotations or rewards. Under this structure, introducing segment-level alarm tokens is also conceptually reasonable.
>
> However, such a triadic design would require careful choices of supervision strategies, temporal credit assignment, and interaction mechanisms between time, vision, and text, along with non-trivial computational overhead. We believe a systematic exploration of these design choices is better left for future work. Temporal grounding would require even finer-grained temporal annotations, which would place additional demands on data quality and annotation cost, and we therefore also consider it an important but separate future extension.

---

### Official Review · Reviewer_k93y · 2025-10-31

**Soundness:** 3
**Presentation:** 2
**Contribution:** 2
**Rating:** 4
**Confidence:** 4

**Summary:**

The paper shows that current video LLMs that look safe in text or image settings often fail once real video is involved, especially when the video itself contains harmful content. To measure this, the authors build VideoSafetyEval, an 11.4k sample benchmark covering 6 risk categories and 19 subcategories in 10 languages, with paired harmful and safe queries so they can test both under defense and over defense. To improve safety they introduce VideoSafetyThinking, a 46k training set with modality level tags, and VideoSafety R1, a two stage alignment method that first trains visual and textual “alarm tokens” to detect harmful signals, then uses safety guided GRPO style RL to reward correct detection and well formatted safe answers.

**Strengths:**

1. This paper makes substantial experiments. Authors evaluate a wide range of publicly available video LLMs in exactly the challenging setting they care about and they do so on a balanced benchmark that separates harmful video plus harmful query (VSE HH), safe video plus harmful query (VSE SH) and safe query for false refusals.
2. This paper is well-written and organized. The presentation of this paper is clear.
3.  The task in this paper focus on safety issue of video LLM, which is important as a large share of the community is now training video LLMs without having a serious safety diagnostic. This paper hands them exactly that diagnostic together with a concrete way to fix at least the current class of failures.

**Weaknesses:**

I'd like to increase my rate if author can address my following concerns:

1. A large part of the safety gains is established with a model in the loop evaluator (Qwen based). The policy which indicates the definition of harmfulness is missing and so the definition of harmfulness is fully dependent on Qwen model, which may limit the future research.
2. The base model used in ablation is unknown. It's possible gain of each component is different across different base model.
3. The paper claims that harmful video semantics, not just more frames, cause the safety drop. But experiments do not systematically sweep frame numbers for the same clip across several models. A controlled frame sweep would make that claim firmer.

**Questions:**

Proposed model may have limit addressing long videos (> 60 sec), as the length of training video mostly are less than 60 sec. It's better to add this in limitation part.

---

> ### Author Response · Authors · 2025-11-22
> **W1: Clarification on Evaluation**
>
> Thank you for highlighting this important point. We understand the concern that relying solely on a model for safety definitions could be limiting. We would like to clarify that in our current setup, the definition of harmfulness is actually grounded in the official policies of mainstream platforms (e.g., TikTok and YouTube), rather than being generated by the Qwen evaluator. In the evaluation stage, **the Qwen-based model is only used to assess whether a system shows a refusal or cautionary attitude toward these policy-defined harmful queries with relevant videos, rather than to redefine what is harmful**. Moreover, we **have verified the consistency between Qwen-based evaluation and human evaluation** in Table 12 of Appendix H.3.
>
> That said, we fully agree with your insight that while this approach suits our current investigation into concrete defenses and attacks, it may be less adequate for complex, abstract safety scenarios. To address this, we have revised the manuscript to explicitly discuss these boundaries and the implications for future research.

---

> ### Author Response · Authors · 2025-11-22
> **W2: Ablations**
>
> Our ablation studies are conducted on VideoLLaMA3-2B, which **serves as our primary base model throughout the paper**. To address the reviewer’s concern about potential variability across different architectures, we additionally perform the same ablations on Qwen2.5 VL-3B, a model with a different backbone and training recipe. The results, shown in the table below, lead to conclusions that are consistent with those observed on VideoLLaMA3-2B. This cross-model agreement suggests that the contribution of each component is stable and not tied to a specific base model. Due to time constraints, we will further refine and complete this ablation.
> | Fine-Tuning | Alarm Token | CLS Task | GRPO | DRA | VSE (DSR) | VSE (HS) | MMBench (DSR) |
> |-------------|-------------|----------|------|-----|-----------|----------|----------------|
> |             |             |          |      |     | 54.0%     | 2.4      | 31.2%          |
> | ✓           |             |          |      |     | 69.1%     | 3.5      | 37.0%          |
> | ✓           | ✓           |          |      |     | 72.3%     | 3.5      | 38.1%          |
> | ✓           | ✓           | ✓        |      |     | 77.9%     | 3.6      | 41.6%          |
> |             |             |          | ✓    | ✓   | 83.7%     | 3.3      | 58.6%          |
> | ✓           | ✓           | ✓        | ✓    |     | 85.3%     | 3.7      | 74.7%          |
> | ✓           | ✓           | ✓        | ✓    | ✓   | 91.7%     | 3.6      | 80.6%          |

---

> ### Author Response · Authors · 2025-11-22
> **W3: Effects of Token Load**
>
> We appreciate the reviewer for offering this constructive suggestion. In the Effects of FPS section, we have already performed a preliminary analysis of changing the number of frames for the same video. As discussed in Section 5.5, however, increasing FPS in realistic long videos typically affects both (i) the semantics that are actually sampled (by extending existing segments or exposing new harmful snippets) and (ii) the token load. To more rigorously support our conclusion that harmful video semantics, not just more token load, cause the safety drop, we decouple these two factors and study them separately.
>
> To analyze token load under fixed semantics, our original experimental design follows the suggestion of reviewer 8QWg (Q2): we construct a static video dataset, VLGuard-Video (Appendix I.2), by extending a single malicious image into a 10-second static video, and then sampling {1, 10} frames to compute DSR for multiple Video-LLMs (Table 3). In this rebuttal, we further extend the sampled frame counts to {1, 10, 20, 30, 40, 50, 60}. As shown in the table below, for the 7B model, the fluctuation has a standard deviation of approximately 1.00%, while for the 3B model, the standard deviation is around 0.66%. When semantics are strictly identical, increasing the token load only causes minor fluctuations in DSR across models and does not lead to any systematic collapse in safety.
>
> In contrast, analyzing harmful semantics under approximately fixed token load is more straightforward. VSE-HH and VSE-SH correspond to harmful multimodal attacks with harmful-video semantics and benign-video semantics, respectively. By evaluating them at the same sampling rate (Table 3), we observe that VSE-HH causes a much larger safety degradation than VSE-SH. Taken together, these experiments support our claim that the observed safety drop is primarily driven by harmful video semantics rather than merely by an increase in token load.
>
> | Model / FPS        | 1      | 10     | 20     | 30     | 40     | 50     | 60     | Std.  |
> |--------------------|:------:|:------:|:------:|:------:|:------:|:------:|:------:|:-----:|
> | **Qwen2.5 VL-7B**      | 45.5% | 44.6% | 43.8% | 45.5% | 45.1% | 46.0% | 47.0% | 1.00% |
> | **VideoLLaMA3-7B**     | 58.6% | 57.4% | 58.0% | 56.1% | 58.8% | 57.7% | 56.2% | 1.07% |
> | **Qwen2.5 VL-3B**      | 52.6% | 52.0% | 52.0% | 52.3% | 51.4% | 51.1% | 50.9% | 0.66% |
> | **VideoLLaMA3-2B**     | 44.9% | 44.5% | 44.8% | 44.6% | 45.3% | 43.4% | 45.4% | 0.66% |

---

> > ### Comment · Reviewer_k93y · 2025-11-26
> >
> > Thanks for your answer to my concerns. For W3, I am curious why larger models like Qwen2.5-VL-7B would have a larger standard deviation than smaller models like Qwen2.5-VL-3B.

---

> ### Author Response · Authors · 2025-11-22
> **Q1: Analysis of Handling Longer Videos**
>
> Thank you for raising this valid concern. We recognize that insufficient exposure to long-duration videos can limit a model's ability to capture long-term dependencies, particularly in perception tasks. At the same time, it is worth noting that for safety purposes, long videos often present distinct challenges, such as multiple hazardous events.
>
> Our observations indicate that safety alignment benefits significantly from semantic diversity. To verify this, we analyzed DSR across various event lengths and found that our models maintain stability even as video length increases, unlike baselines. We fully agree, however, that the current duration cap is a constraint worth highlighting. We have revised the limitation section to openly discuss the implications of training on shorter clips and the potential value of incorporating long-form videos in future research.
>
> | Model                         | 1–60 sec. | 61–120 sec. | 121–360 sec. |
> |------------------------------|:---------:|:-----------:|:------------:|
> | **VideoLLaMA3-2B**              |  19.4%    |    16.1%    |    15.6%     |
> | **Qwen2.5 VL-3B**            |  56.1%    |    50.0%    |    47.9%     |
> | **VideoLLaMA3-2B + VideoSafetyR1**   |  89.1%    |    89.9%    |    91.4%     |
> | **Qwen2.5 VL-3B + VideoSafetyR1** |  92.0%    |    90.6%    |    90.8%     |

---

> ### Author Response · Authors · 2025-11-26
>
> We thank the reviewer for the helpful follow-up. The difference in standard deviation reflects differences in how sensitive the models are to changes in the input distribution near the task difficulty boundary. Prior work has examined how LLMs can be sensitive to modifications of the prompt format[1][2][3]. In particular, [4] reports that within the Qwen family smaller models tend to produce more stable responses. In our setting, changing the number of frames also perturbs the input distribution and leads to changes in the model outputs. Our results are consistent with [4] and show that for Qwen based language models the smaller model exhibits higher response stability than the larger one.
>
> We also emphasize that the absolute values of the standard deviations are very small. This indicates that the overall sensitivity to changes in frame count is limited. **The numerical differences in standard deviation between models are also minor, which suggests that the difference in sensitivity across models is modest.**
>
> If you have any further questions, please feel free to let us know. We once again sincerely thank you for your time and consideration.
>
> **Ref:**
>
> [1] QUANTIFYING LANGUAGE MODELS' SENSITIVITY TOSPURIOUS FEATURES IN PROMPT DESIGN or: How I learned to start worrying about prompt formatting
>
> [2] Promptception: How Sensitive Are Large Multimodal Models to Prompts?
>
> [3] Benchmarking Prompt Sensitivity in Large Language Models
>
> [4] Prompt Stability in Code LLMs: Measuring Sensitivity across Emotion- and Personality-Driven Variations

---

> > ### Comment · Reviewer_k93y · 2025-11-26
> >
> > Thanks for addressing my concerns. I'd like to increase my score.

---

> > > ### Author Response · Authors · 2025-11-27
> > > **Sincere Appreciation for Your Encouraging Evaluation**
> > >
> > > We sincerely appreciate your positive evaluation and the improved score, which is greatly encouraging to us. We are also deeply grateful for your valuable suggestions, which have helped us further refine and strengthen the research presented in this work.

---

### Author Response · Authors · 2025-11-22
**Author's Rebuttal**

We sincerely appreciate the thoughtful and insightful feedback from all four reviewers. Their comments have rightly highlighted two central aspects of our work, the design of the data annotation process involving Qwen-in-the-loop, and the overall rationale behind the SFT+RL pipeline, which are indeed critical to this study. **In response, we have carefully reply each of the reviewers’ concerns point by point in the following comments.** We will promptly submit a revised manuscript and remain fully prepared to incorporate any additional experiments.

---

### Author Response · Authors · 2025-11-25
**Revision Details**

We sincerely appreciate the reviewers' valuable suggestions. We have now submitted a revised version of the manuscript based on their feedback, with all modifications highlighted in blue. **This version presents our revision plan, and we will further refine the details in the final manuscript. We also welcome any additional suggestions from the reviewers.** Below, we provide a brief summary of the changes made in the order of their occurrence in the manuscript:

1. Based on Reviewer _XGNf_'s _W1_ and Reviewer _JMVT_'s _W1_, we have revised the statement **"first large-scale real-world dataset."**

2. In response to Reviewer _XGNf_'s _W1_, _Q1_, and _Q2_, we added a description and comparison of the VAD task in **Related Work**, and the comparison experiment with HolmesVAD is now included in **Appendix O**.

3. Following Reviewer _XGNf_'s _W4_, we shortened **Section 3** to half a page, with details moved to **Appendix C**.

4. Based on Reviewer _XGNf_'s _W1_, we added the conclusion and experiment showing that fine-tuning does not necessarily reduce the safety of language models, as discussed in **Effects of Multimodal Fine-tuning** and shown in **Figure 5**.

5. In response to Reviewer _JMVT_'s _Q2_, we included VideoHallucer experiments in **Generalization on Other Video Benchmarks** and **Table 3**.

6. Based on Reviewer _k93y_'s _Q1_, we added an analysis of the relationship between video length and DSR, presented in **Generalization on Video Length** and **Table 4**.

7. In response to Reviewer _k93y_'s _W3_, Reviewer _8QWg_'s _W2_ and _Q2_, and Reviewer _XGNf_'s _W3.1_ and _W3.3_, we refined the VLGuard-Video experiments and explanations in **Figure 6**.
   Additionally, we moved the thumbnail-based harmfulness analysis from **Appendix J** to **Section 5.5** and **Table 5**, with example figures provided in **Appendix K**.

8. Based on Reviewer _XGNf_'s _W2_ and _W3.2_, we added an Attention Sink analysis and explanations of SFT and RL in **Section 5.6** and **Figure 7**.

9. We supplemented the **Limitation** section with additional content.

10. Following Reviewer _JMVT_'s _W2_, we added experiments on human validation in **Appendix E.2**.

11. Based on Reviewer _JMVT_'s _Q1_, we included hyperparameter descriptions in **Appendix H**.

12. In response to Reviewer _JMVT_'s _Q3_, we added inference diversity analysis in **Appendix L** and multilingual DSR analysis in **Appendix N**.

13. According to Reviewer _k93y_'s _W2_, we added an ablation study of Qwen2.5 VL-3B in **Appendix M**.

14. Following Reviewer _8QWg_'s _W1_, we refined and improved the examples shown in **Appendix Q: Cases**.

---

### Author Response · Authors · 2025-12-01
**Summary (Part II)**

# Responses to *XGNf*
| Issue ID | Reviewer Question | Our Response & Outcome |
|----------|-------------------|-------------------------|
| W1 | (1) Missing citations of Video Anomaly Detection (VAD) papers.  (2) Claim that fine-tuning harms safety is a common consensus. | (1) Safety alignment and VAD focus on fundamentally different objectives, and XGNf appears to conflate these two tasks. We provided a detailed comparison in our response and added a VAD overview to the Related Work section.  (2) This “consensus” is an intuitive assumption rather than an established conclusion in Safety Alignment. We supplemented and challenged this claim with empirical evidence, and added the corresponding experiment to Figure 5. |
| W2 | Analyze the advantages of AT-SFT and GRPO | (1) We explained the potential mechanism of AT-SFT through the Attention Sink phenomenon—specifically, how releasing redundant visual sink tokens into text sink tokens improves alignment; detailed experiments are added in Section 5.6.  (2) By comparing PPO and DPO, we explained why GRPO offers advantages under a chain-of-thought training framework. |
| W3.1 | Whether the FPS drop is related to exceeding the model’s token understanding limits | This consideration was already addressed in the initial version and explained in Appendix C (Appendix D in the latest version). In fact, the sampled number of frames is within the model’s comprehension range.|
| W3.2 | Challenge to the claim that “RL is a cure for safety” | We found this concern somewhat confusing. In the paragraph “L409–L414” mentioned by reviewer XGNf, we did **not** claim that “RL is a cure for safety.” That section discusses that RL-based models demonstrate lower refusal rates (a metric for over-defensive behavior), reflecting RL’s generalization capability. Other experiments also support this interpretation. See discussion for detailed explanation. |
| W3.3 | Questioning the role of static videos | We explained the rationale for including static videos. This was already covered in Section 5.5 and Appendix I.2 (K.2 in the latest version) of the initial submission. |
| W4 | Reduce the length of Section 3 | Following XGNf’s suggestion, we moved the detailed content to Appendix C. |
| Q1 | Comparative analysis with SafeWatch | We discussed the differences in motivation, mechanism, and architecture based on the fundamental distinction in task objectives. |
| Q2 | Performance comparison with Holmes-VAD | We conducted experiments on both safety-alignment benchmarks and VAD benchmarks. Each model excels at its own task but performs modestly on the other task, further validating the fundamental task differences. |
# Responses to *JMVT*
| Issue ID | Reviewer Question | Our Response & Outcome |
|----------|-------------------|-------------------------|
| Q1 | Analyze the effects of λ₁ and λ₂ | We added ablation studies for λ₁ and λ₂, showing that their specific values do not affect the effectiveness of AT-SFT. The detailed hyperparameter settings have been included in Appendix H. |
| Q2 | Analyze general video understanding capability | (1) This part was already included in Table 3 of the initial version.  (2) We additionally incorporated the results of VideoHallucer into Table 3. |
| Q3 | Analyze reasoning diversity and the impact of multilingual prompts | We conducted a detailed analysis, and the results have been added to Appendix L and Appendix N. |

---

### Author Response · Authors · 2025-12-01
**Summary (Part I)**

# Summary of Discussion Outcomes
After our in-depth discussion with *k93y* (initial score: 4), the reviewer acknowledged our responses and **raised the score to 6**.
Meanwhile, *JMVT* (initial score: 6) agreed that our clarifications addressed their concerns and expressed willingness to **increase the score**.
*XGNf* (initial score: 2) and *8QWg* (initial score: 8) did not participate in the discussion before closing the comment window.

*We summarized the key questions raised by each reviewer and our corresponding responses; please refer to the detailed replies provided under each reviewer.*

# Shared Questions
| Issue ID | Reviewer Concern | Our Response & Outcome |
|----------|------------------|-------------------------|
| JMVT's W1, XGNf's W1 | The phrase “the first large-scale, real-world dataset” was considered overstated | We revised the wording in the latest version, which has been acknowledged and accepted by JMVT. |
| k93y's W1, JMVT's W2 | Verification of the dataset construction pipeline and evaluation reliability. | (1) We had already validated the evaluation reliability in Appendix H.3 of the initial submission (Appendix J.3 in the latest version). (2) During discussion, we additionally hired a professional annotation team for further verification, and the results were acknowledged by both k93y and JMVT. |
| k93y's W3, 8QWg's W2 & Q2 | Token load analysis | The experiment in our initial submission aligns with the analysis proposed by 8QWg in Q2. After adding further experimental results, our analysis received approval from k93y. |

# Responses to *k93y*
| Issue ID | Reviewer Question | Our Response & Outcome |
|----------|-------------------|-------------------------|
| W2 | Is the module effective on other models? | We added an ablation study on Qwen2.5 VL-3B, which consistently confirms the module’s effectiveness. This experiment has been included in Appendix M. |
| Q1 | Provide additional analysis on the limitations regarding long videos | (1) We validated that our method generalizes to long videos (60s–360s). This experiment is added in Section 5.3 and Table 4.  (2) In Appendix B, we further analyze the safety response limitations for video–query pairs exceeding 360 seconds. |
| Follow-up | Why does the 7B model exhibit larger performance fluctuations than the 3B model as token count increases? | We provided an explanation based on existing research. |

# Responses to *8QWg*
| Issue ID | Reviewer Question | Our Response & Outcome |
|----------|-------------------|-------------------------|
| W1 | Does the dataset lack two types of implicit attack scenarios? | (1) In Appendix Q, we refined the case analysis to demonstrate that VSE contains the first type of implicit attack scenario and a small portion of the second type.   (2) The limitation regarding the insufficient coverage of the second implicit tool-use scenario has been added to Appendix B. |
| Q1 | Provide additional discussion on the design of alarm tokens and reward mechanisms | We further examined the feasibility of the approaches referenced in Q1, and the corresponding discussion has been added to Appendix B. |

---

### Meta-Review · Area_Chair_gTFh · 2026-01-08

**Summary:**

The reviewers agree that this paper makes a strong and timely contribution by introducing the first comprehensive benchmark and a principled defense pipeline for Video LLM safety, clearly demonstrating that video modalities substantially exacerbate safety risks and proposing an effective, well-validated alignment framework to address them.

**Reviewer Concerns:**

nitial concerns regarding overstatement of novelty, evaluation reliability, token-load confounds, dependence on model-based annotation, long-video generalization, and comparison with prior safety and VAD methods were carefully and convincingly addressed through revised claims, additional controlled experiments, cross-model ablations, human validation, clearer positioning against related work, and expanded limitation discussions; no major concerns remain outstanding.

**Reviewer Scores:**

Reviewers who engaged in discussion (k93y, JMVT) explicitly raised their scores after rebuttal. 8QWg (initial score: 8) did not participate in the discussion and should remain positive at this stage.

The rebuttal directly addressed XGNf’s core technical objections with concrete additions—revised “first” claims, stronger related-work positioning (incl. VAD distinctions), new controlled token-load experiments, and expanded comparisons/ablations—rather than just narrative clarification. It also materially reduced their stated uncertainty by adding further empirical evidence (e.g., human validation, additional benchmark comparisons) and tightening presentation/structure. Therefore it's reasonable to expect that initially negative reviewer XGNf would also substantially increase their scores if fully participating.

---

### Decision · Program_Chairs · 2026-01-26

Accept (Poster)